# There Is No Turning Back:
# A Self-Supervised Approach for
# Reversibility-Aware Reinforcement Learning

**Nathan Grinsztajn**[*]
Inria, Scool Team
CRIStAL, CNRS, Université de Lille
`nathan.grinsztajn@inria.fr`

**Johan Ferret**[*]
Google Research, Brain Team
Inria, Scool Team
CRIStAL, CNRS, Université de Lille

**Olivier Pietquin**
Google Research, Brain Team

**Philippe Preux**
Inria, Scool Team
CRIStAL, CNRS, Université de Lille

**Matthieu Geist**
Google Research, Brain Team

## Abstract

We propose to learn to distinguish reversible from irreversible actions for better informed decision-making in Reinforcement Learning (RL). From theoretical considerations, we show that approximate reversibility can be learned through a simple surrogate task: ranking randomly sampled trajectory events in chronological order. Intuitively, pairs of events that are always observed in the same order are likely to be separated by an irreversible sequence of actions. Conveniently, learning the temporal order of events can be done in a fully self-supervised way, which we use to estimate the reversibility of actions from experience, without any priors. We propose two different strategies that incorporate reversibility in RL agents, one strategy for exploration (RAE) and one strategy for control (RAC). We demonstrate the potential of reversibility-aware agents in several environments, including the challenging Sokoban game. In synthetic tasks, we show that we can learn control policies that never fail and reduce to zero the side-effects of interactions, even without access to the reward function.

## 1   Introduction

We address the problem of estimating if and how easily actions can be reversed in the Reinforcement Learning (RL) context. Irreversible outcomes are often not to be taken lightly when making decisions. As humans, we spend more time evaluating the outcomes of our actions when we know they are irreversible [29]. As such, irreversibility can be positive (*i.e.* takes risk away for good) or negative (*i.e.* leads to later regret). Also, decision-makers are more likely to anticipate regret for hard-to-reverse decisions [50]. All in all, irreversibility seems to be a good prior to exploit for more principled decision-making. In this work, we explore the option of using irreversibility to guide decision-making and confirm the following assertion: by estimating and factoring reversibility in the action selection process, safer behaviors emerge in environments with intrinsic risk factors. In addition to this, we

---

[*]Equal contribution.

35th Conference on Neural Information Processing Systems (NeurIPS 2021).

show that exploiting reversibility leads to more efficient exploration in environments with undesirable irreversible behaviors, including the famously difficult Sokoban puzzle game.

However, estimating the reversibility of actions is no easy feat. It seemingly requires a combination of planning and causal reasoning in large dimensional spaces. We instead opt for another, simpler approach (see Fig. 1): we propose to learn in which direction time flows between two observations, directly from the agents' experience, and then consider *irreversible* the transitions that are assigned a temporal direction with high confidence. *In fine*, we reduce reversibility to a simple classification task that consists in predicting the temporal order of events.

Our contributions are the following: 1) we formalize the link between reversibility and precedence estimation, and show that reversibility can be approximated via temporal order, 2) we propose a practical algorithm to learn temporal order in a self-supervised way, through simple binary classification using sampled pairs of observations from trajectories, 3) we propose two novel exploration and control strategies that incorporate reversibility, and study their practical use for directed exploration and safe RL, illustrating their relative merits in synthetic as well as more involved tasks such as Sokoban puzzles.

## 2   Related Work

To the best of our knowledge, this work is the first to explicitly model the reversibility of transitions and actions in the context of RL, using temporal ordering to learn from trajectories in a self-supervised way, in order to guide exploration and control. Yet, several aspects of the problem we tackle were studied in different contexts, with other motivations; we review these here.

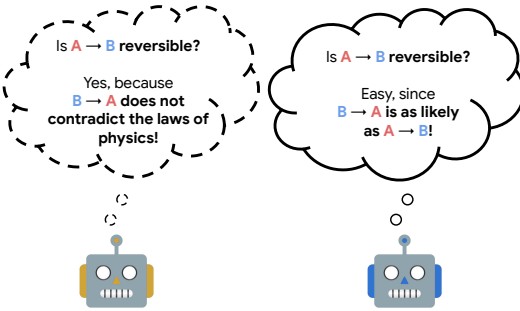

**Leveraging reversibility in RL.** Kruusmaa et al. [26] estimate the reversibility of state-action couples so that robots avoid performing irreversible actions, since they are more likely to damage the robot itself or its environment. A shortcoming of their approach is that they need

Figure 1: High-level illustration of how reversibility can be estimated. **Left:** from an understanding of physics. **Right:** ours, from experience.

to collect explicit state-action pairs and their reversal actions, which makes it hard to scale to large environments. Several works [40, 5, 4] use reachability as a curiosity bonus for exploration: if the current state has a large estimated distance to previous states, it means that it is novel and the agent should be rewarded. Reachability and reversibility are related, in the sense that irreversible actions lead to states from which previous states are unreachable. Nevertheless, their motivations and ours diverge, and we learn reversibility through a less involved task than that of learning reachability. Nair et al. [33] learn to reverse trajectories that start from a goal state so as to generate realistic trajectories that reach similar goals. In contrast, we use reversibility to direct exploration and/or control, not for generating learning data. Closest to our work, Rahaman et al. [37] propose to learn a potential function of the states that increases with time, which can detect irreversibility to some extent. A drawback of the approach is that the potential function is learned using trajectories sampled from a random policy, which is a problem for many tasks where a random agent might fail to cover interesting parts of the state space. In comparison, our method does not use a potential function and learns jointly with the RL agent, which makes it a viable candidate for more complex tasks.

**Safe exploration.**   Safe exploration aims at making sure that the actions of RL agents do not lead to negative or unrecoverable effects that would outweigh the long-term value of exploration [2]. Notably, previous works developed distinct approaches to avoid irreversible behavior: by incremental updates to safe policies [23, 18], which requires knowing such a policy in advance; by restricting policy search to ergodic policies [32] (*i.e.* that can always come back to any state visited), which is costly; by active exploration [28], where the learner can ask for rollouts instead of exploring potentially unsafe areas of the state space itself; and by computing regions of attraction [9] (the part of the state space where a controller can bring the system back to an equilibrium point), which requires prior knowledge of the environment dynamics.

**Self-supervision from the arrow of time.** Self-supervision has become a central component of modern machine learning algorithms, be it for computer vision, natural language or signal processing. In particular, using temporal consistency as a source of self-supervision is now ubiquitous, be it to learn representations for downstream tasks [19, 38, 12], or to learn to detect temporal inconsistencies [47]. The closest analogies to our work are methods that specifically estimate some aspects of the arrow of time as self-supervision. Most are to be found in the video processing literature, and self-supervised tasks include predicting which way the time flows [35, 47], verifying the temporal order of a subset of frames [30], predicting which video clip has the wrong temporal order among a subset [17] as well as reordering shuffled frames or clips from the video [16, 14, 48]. Bai et al. [6] notably propose to combine several of these pretext tasks along with data augmentation for video classification. Using time as a means of supervision was also explored for image sequencing [8], audio [11] or EEG processing [39]. In RL, self-supervision also gained momentum in recent years [22, 44, 49], with temporal information being featured [1]. Notably, several works [3, 13, 21, 43] leverage temporal consistency to learn useful representations, effectively learning to discriminate between observations that are temporally close and observations that are temporally distant. In comparison to all these works, we estimate the arrow of time through temporal order prediction with the explicit goal of finding irreversible transitions or actions.

## 3    Reversibility

**Degree of Reversibility.** We start by introducing formally the notion of reversibility. Intuitively, an action is reversible if it can be undone, meaning that there is a sequence of actions that can bring us back to the original state.

**Definition 1.** *Given a state $s$, we call* degree of reversibility within $K$ steps *of an action $a$*

$$\phi_K(s, a) := \sup_\pi p_\pi(s \in \tau_{t+1:t+K+1} \mid s_t = s, a_t = a),$$

*and the* degree of reversibility *of an action is defined as*

$$\phi(s, a) := \sup_\pi p_\pi(s \in \tau_{t+1:\infty} \mid s_t = s, a_t = a),$$

*with $\tau = \{s_i\}_{i=1\ldots T} \sim \pi$ corresponding to a trajectory, and $\tau_{t:t'}$ the subset of the trajectory between the timesteps $t$ and $t'$ (excluded). We omit their dependency on $\pi$ for the sake of conciseness. Given $s \in S$, the action $a$ is* reversible *if and only if $\phi(s, a) = 1$, and said* irreversible *if and only if $\phi(s, a) = 0$.*

In deterministic environments, an action is either reversible or irreversible: given a state-action couple $(s, a)$ and the unique resulting state $s'$, $\phi_K(s, a)$ is equal to 1 if there is a sequence of less than $K$ actions which brings the agent from $s'$ to $s$, and is otherwise equal to zero. In stochastic environments, a given sequence of actions can only reverse a transition up to some probability, hence the need for the notion of degree of reversibility.

**Policy-Dependent Reversibility.** In practice, it is useful to quantify the degree of reversibility of an action as the agent acts according to a fixed policy $\pi$, for which we extend the notions introduced above. We simply write :

$$\phi_{\pi,K}(s, a) := p_\pi(s \in \tau_{t+1:t+K+1} \mid s_t = s, a_t = a) \text{ and } \phi_\pi(s, a) := p_\pi(s \in \tau_{t+1:\infty} \mid s_t = s, a_t = a).$$

It immediately follows that $\phi_K(s, a) = \sup_\pi \phi_{\pi,K}(s, a)$ and $\phi(s, a) = \sup_\pi \phi_\pi(s, a)$.

## 4    Reversibility Estimation via Classification

Quantifying the exact degree of reversibility of actions is generally hard. In this section, we show that reversibility can be approximated efficiently using simple binary classification.

### 4.1    Precedence Estimation

Supposing that a trajectory contains the states $s$ and $s'$, we want to be able to establish *precedence*, that is predicting whether $s$ or $s'$ comes first *on average*. It is a binary classification problem, which

consists in estimating the quantity $\mathbb{E}_{s_t=s,s_{t'}=s'}\big[\mathbb{1}_{t'>t}\big]$. Accordingly, we introduce the precedence estimator which, using a set of trajectories, learns to predict which state of an arbitrary pair is most likely to come first.

**Definition 2.** *Given a fixed policy $\pi$, we define the* finite-horizon precedence estimator *between two states as follows:*

$$\psi_{\pi,T}(s,s') = \mathbb{E}_{\tau\sim\pi}\,\mathbb{E}_{\substack{s_t=s,s_{t'}=s'\\t,t'<T}}\big[\mathbb{1}_{t'>t}\big].$$

Conceptually, given two states $s$ and $s'$, the precedence estimator gives an approximate probability of $s'$ being visited after $s$, given that both $s$ and $s'$ are observed in a trajectory. The indices are sampled uniformly within the specified horizon $T \in \mathbb{N}$, so that this quantity is well-defined even for infinite trajectories. Additional properties of $\psi$, regarding transitivity for instance, can be found in Appx. A.2.

**Remark 1.** *The quantity $\psi_{\pi,T}(s,s')$ is only defined for pairs of states which can be found in the same trajectory, and is otherwise irrelevant. In what follows, we implicitly impose this condition when considering state pairs.*

**Theorem 1.** *For every policy $\pi$ and $s,s' \in S$, $\psi_{\pi,T}(s,s')$ converges when $T$ goes to infinity. We refer to the limit as the* precedence estimator, *written $\psi_\pi(s,s')$.*

The proof of this theorem is developed in Appendix A.3. This result is key to ground theoretically the notion of empirical reversibility $\bar{\phi}$, which we introduce in the next definition. It simply consists in extending the notion of precedence to a state-action pair.

**Definition 3.** *We finally define the* empirical reversibility *using the precedence estimator:*

$$\bar{\phi}_\pi(s,a) = \mathbb{E}_{s'\sim P(s,a)}\big[\psi_\pi(s',s)\big].$$

In a nutshell, given that we start in $s$ and take the action $a$, the empirical reversibility $\bar{\phi}_\pi(s,a)$ measures the probability that we go back to $s$, starting from a state $s'$ that follows $(s,a)$. We now show that our empirical reversibility is linked with the notion of reversibility defined in the previous section, and can behave as a useful proxy.

## 4.2 Estimating Reversibility from Precedence

We present here our main theoretical result which relates reversibility and empirical reversibility:

**Theorem 2.** *Given a policy $\pi$, a state $s$ and an action $a$, we have: $\bar{\phi}_\pi(s,a) \geq \frac{\phi_\pi(s,a)}{2}$.*

The full proof of the theorem is given in Appendix A.3.

This result theoretically justifies the name of empirical reversibility. From a practical perspective, it provides a way of using $\bar{\phi}$ to detect actions which are irreversible or hardly reversible: $\bar{\phi}_\pi(s,a) \ll 1$ implies $\phi_\pi(s,a) \ll 1$ and thus provides a sufficient condition to detect actions with low degrees of reversibility. This result gives a way to detect actions that are irreversible given a specific policy followed by the agent. Nevertheless, we are generally interested in knowing if these actions are irreversible for any policy, meaning $\phi(s,a) \ll 1$ with the definition of Section 3. The next proposition makes an explicit connection between $\bar{\phi}_\pi$ and $\phi$, under the assumption that the policy $\pi$ is stochastic.

**Proposition 1.** *We suppose that we are given a state $s$, an action $a$ such that $a$ is reversible in $K$ steps, and a policy $\pi$. Under the assumption that $\pi$ is stochastic enough, meaning that there exists $\rho > 0$ such that for every state and action $s,a$, $\pi(a \mid s) > \rho$, we have: $\bar{\phi}_\pi(s,a) \geq \frac{\rho^K}{2}$. Moreover, we have for all $K \in \mathbb{N}$: $\bar{\phi}_\pi(s,a) \geq \frac{\rho^K}{2}\phi_K(s,a)$.*

The proof is given in Appendix A.4. As before, this proposition gives a practical way of detecting irreversible moves. If for example $\bar{\phi}_\pi(s,a) < \rho^k/2$ for some $k \in \mathbb{N}$, we can be sure that action $a$ is not reversible in $k$ steps. The quantity $\rho$ can be understood as a minimal probability of taking any action in any state. This condition is not very restrictive: $\epsilon$-greedy strategies for example satisfy this hypothesis with $\rho = \frac{\epsilon}{|A|}$.

In practice, it can also be useful to limit the maximum number of time steps between two sampled states. That is why we also define the windowed precedence estimator as follows:

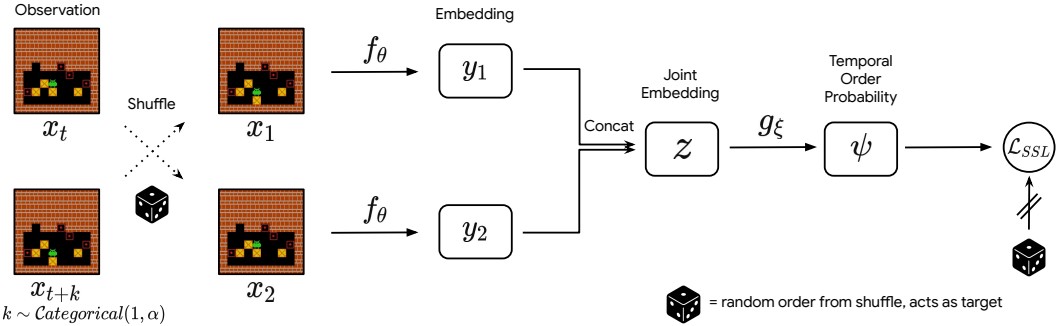

Figure 2: The proposed self-supervised procedure for precedence estimation.

**Definition 4.** *Given a fixed policy $\pi$, we define the* windowed precedence estimator *between two states as follows:*

$$\psi_{\pi,T,w}(s,s') = \mathbb{E}_{\tau\sim\pi}\mathbb{E}_{\substack{s_t=s,s_{t'}=s' \\ t,t'<T \\ |t-t'|\leq w}}\left[\mathbb{1}_{t'>t}\right].$$

Intuitively, compared to previous precedence estimators, $\psi_{\pi,T,w}$ is restricted to short-term dynamics, which is a desirable property in tasks where distinguishing the far future from the present is either trivial or impossible.

## 5 Reversibility-Aware Reinforcement Learning

Leveraging the theoretically-grounded bridge between precedence and reversibility established in the previous section, we now explain how reversibility can be learned from the agent's experience and used in a practical setting.

**Learning to rank events chronologically.**   Learning which observation comes first in a trajectory is achieved by binary supervised classification, from pairs of observations sampled uniformly in a sliding window on observed trajectories. This can be done fully offline, *i.e.* using a previously collected dataset of trajectories for instance, or fully online, *i.e.* jointly with the learning of the RL agent; but also anywhere on the spectrum by leveraging variable amounts of offline and online data.

This procedure is not without caveats. In particular, we want to avoid overfitting to the particularities of the behavior of the agent, so that we can learn meaningful, generalizable statistics about the order of events in the task at hand. Indeed, if an agent always visits the state $s_a$ before $s_b$, the classifier will probably assign a close-to-one probability that $s_a$ precedes $s_b$. This might not be accurate with other agents equipped with different policies, unless transitioning from $s_b$ to $s_a$ is hard due to the dynamics of the environment, which is in fact exactly the cases we want to uncover. We make several assumptions about the agents we apply our method to: 1) agents are learning and thus, have a policy that changes through interactions in the environment, 2) agents have an incentive not to be too deterministic. For this second assumption, we typically use an entropic regularization in the chosen RL loss, which is a common design choice in modern RL methods. These assumptions, when put together, alleviate the risk of overfitting to the idiosyncrasies of a single, non-representative policy.

We illustrate the precedence classification procedure in Fig. 2. A temporally-ordered pair of observations, distant of no more than $w$ timesteps, is sampled from a trajectory and uniformly shuffled. The result of the shuffling operation is memorized and used as a target for the binary classification task. A Siamese network creates separate embeddings for the pair of observations, which are concatenated and fed to a separate feed-forward network, whose output is passed through a sigmoid to obtain a probability of precedence. This probability is updated via negative log-likelihood against the result of the shuffle, so that it matches the actual temporal order.

Then, a transition (and its implicit sequence of actions) represented by a starting observation $x$ and a resulting observation $x'$ is deemed irreversible if the estimated precedence probability $\psi(x,x')$ is superior to a chosen threshold $\beta$. Note that we do not have to take into account the temporal proximity of these two observations here, which is a by-product of sampling observations uniformly

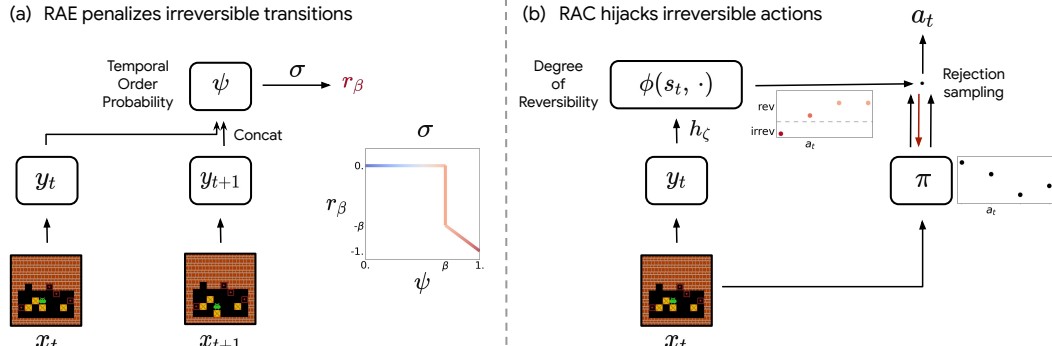

Figure 3: Our proposed methods for reversibility-aware RL. **(a):** RAE encourages reversible behavior via auxiliary rewards. **(b):** RAC avoids irreversible behavior by rejecting actions whose estimated reversibility is inferior to a threshold.

in a window in trajectories. Also, depending on the threshold $\beta$, we cover a wide range of scenarios, from pure irreversibility ($\beta$ close to 1) to soft irreversibility ($\beta > 0.5$, the bigger $\beta$, the harder the transition is to reverse). This is useful because different tasks call for different levels of tolerance for irreversible behavior: while a robot getting stuck and leading to an early experiment failure is to be avoided when possible, tasks involving human safety might call for absolute zero tolerance for irreversible decision-making. We elaborate on these aspects in Sec. 6.

**Reversibility-Aware Exploration and Control.** We propose two different algorithms based on reversibility estimation: Reversibility-Aware Exploration (RAE) and Reversibility-Aware Control (RAC). We give a high-level representation of how the two methods operate in Fig. 3.

In a nutshell, RAE consists in using the estimated reversibility of a pair of consecutive observations to create an auxiliary reward function. In our experiments, the reward function is a piecewise linear function of the estimated reversibility and a fixed threshold, as in Fig. 3: it grants the agent a negative reward if the transition is deemed too hard to reverse. The agent optimizes the sum of the extrinsic and auxiliary rewards. Note that the specific function we use penalizes irreversible transitions but could encourage such transitions instead, if the task calls for it.

RAC can be seen as the action-conditioned counterpart of RAE. From a single observation, RAC estimates the degree of reversibility of all available actions, and "takes control" if the action sampled from the policy is not reversible enough (*i.e.* has a reversibility inferior to a threshold $\beta$). "Taking control" can have many forms. In practice, we opt for rejection sampling: we sample from the policy until an action that is reversible enough is sampled. This strategy has the advantage of avoiding irreversible actions entirely, while trading-off pure reversibility for performance when possible. RAC is more involved than RAE, since the action-conditioned reversibility is learned from the supervision of a standard, also learned precedence estimator. Nevertheless, our experiments show that it is possible to learn both estimators jointly, at the cost of little overhead.

We now discuss the relative merits of the two methods. In terms of applications, we argue that RAE is more suitable for directed exploration, as it only encourages reversible behavior. As a result, irreversible behavior is permitted if the benefits (*i.e.* rewards) outweigh the costs (*i.e.* irreversibility penalties). In contrast, RAC shines in safety-first, real-world scenarios, where irreversible behavior is to be banned entirely. With an optimal precedence estimator and task-dependent threshold, RAC will indeed hijack all irreversible sampled actions. RAC can be especially effective when pre-trained on offline trajectories: it is then possible to generate fully-reversible, safe behavior from the very first online interaction in the environment. We explore these possibilities experimentally in Sec. 6.2.

Both algorithms can be used online or offline with small modifications to their overall logic. The pseudo-code for the online version of RAE and RAC can be found in Appendix B.2.

The self-supervised precedence classification task could have applications beyond estimating the reversibility of actions: it could be used as a means of getting additional learning signal or representational priors for the RL algorithm. Nevertheless, we opt for a clear separation between the

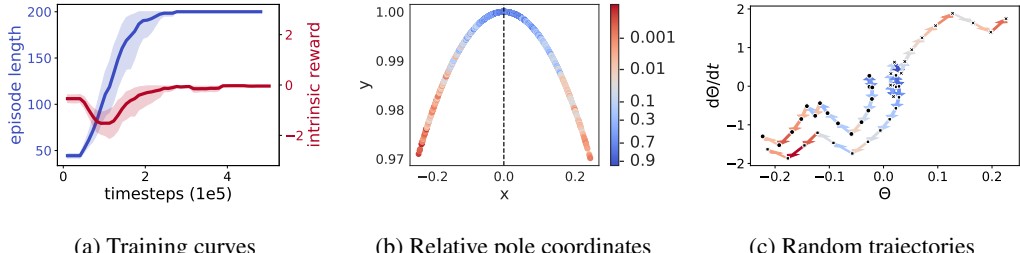

|   |   |   |
|---|---|---|
| (a) Training curves | (b) Relative pole coordinates | (c) Random trajectories |

Figure 4: **(a):** Training curves of a PPO+RAE agent in reward-free Cartpole. Blue: episode length. Red: intrinsic reward. A 95% confidence interval over 10 random seeds is shown. **(b):** The $x$ and $y$ axes are the coordinates of the end of the pole relatively to the cart position. The color denotes the online reversibility estimation between two consecutive states (logit scale). **(c):** The representation of three random trajectories according to $\theta$ (angle of the pole) and $\frac{d\theta}{dt}$. Arrows are colored according to the learned reversibility of the transitions they correspond to.

reversibility and the RL components so that we can precisely attribute improvements to the former, and leave aforementioned studies for future work.

## 6 Experiments

The following experiments aim at demonstrating that the estimated precedence $\psi$ is a good proxy for reversibility, and at illustrating how beneficial reversibility can be in various practical cases. We benchmark RAE and RAC on a diverse set of environments, with various types of observations (tabular, pixel-based), using neural networks for function approximation. See Appendix C for details.

### 6.1 Reward-Free Reinforcement Learning

We illustrate the ability of RAE to learn sensible policies without access to rewards. We use the classic pole balancing task Cartpole [7], using the OpenAI Gym [10] implementation. In the usual setting, the agent gets a reward of 1 at every time step, such that the total undiscounted episode reward is equal to the episode length, and incentivizes the agent to learn a policy that stabilizes the pole. Here, instead, we remove this reward signal and give a PPO agent [42] an intrinsic reward based on the estimated reversibility, which is learned online from agent trajectories. The reward function penalizes irreversibility, as shown in Fig. 3. Note that creating insightful rewards is quite difficult: too frequent negative rewards could lead the agent to try and terminate the episode as soon as possible.

We display our results in Fig. 4. Fig. 4a confirms the claim that RAE can be used to learn meaningful rewards. Looking at the intrinsic reward, we discern three phases. Initially, both the policy and the reversibility classifier are untrained (and intrinsic rewards are 0). In the second phase, the classifier is fully trained but the agent still explores randomly (intrinsic rewards become negative). Finally, the agent adapts its behavior to avoid penalties (intrinsic rewards go to 0, and the length of trajectories increases). Our reward-free agent reaches the score of 200, which is the highest possible score.

To further assess the quality of the learned reversibility, we freeze the classifier after 300k timesteps and display its predicted probabilities according to the relative coordinates of the end of the pole (Fig. 4b) and the dynamics of the angle of the pole $\theta$ (Fig. 4c). In both cases, the empirical reversibility matches our intuition: the reversibility should decrease as the angle or angular momentum increase, since these coincide with an increasing difficulty to go back to the equilibrium.

### 6.2 Learning Reversible Policies

In this section, we investigate how RAE can be used to learn reversible policies. When we train an agent to achieve a goal, we usually want it to achieve that goal following implicit safety constraints. Handcrafting such safety constraints would be time-consuming, difficult to scale for complex problems, and might lead to reward hacking; so a reasonable proxy consists in limiting irreversible side-effects in the environment [27].

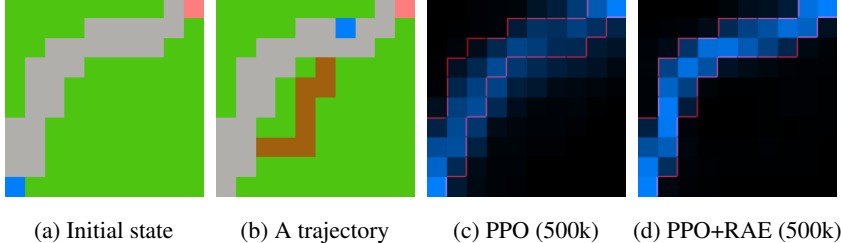

| (a) Initial state | (b) A trajectory | (c) PPO (500k) | (d) PPO+RAE (500k) |

Figure 5: **(a):** The Turf environment. The agent can walk on grass, but the grass then turns brown. **(b):** An illustrative trajectory where the agent stepped on grass pixels. **(c):** State visitation heatmap for PPO. **(d):** State visitation heatmap for PPO+RAE. It coincides with the stone path (red).

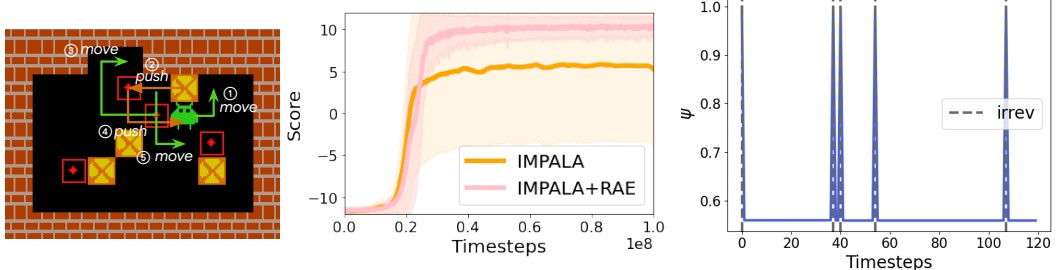

Figure 6: **(a):** Non-trivial reversibility: pushing the box against the wall can be reversed by pushing it to the left, going around, pushing it down and going back to start. A minimum of 17 moves is required to go back to the starting state. **(b):** Performances of IMPALA and IMPALA+RAE on 1k levels of Sokoban (5 seeds average). **(c):** Evolution of the estimated reversibility along one episode.

To quantify side-effects, we propose Turf, a new synthetic environment. As depicted in Fig. 5a,5b, the agent (blue) is rewarded when reaching the goal (pink). Stepping on grass (green) will spoil it, causing it to turn brown. Stepping on the stone path (grey) does not induce any side-effect.

In Fig. 5c,5d, we compare the behaviors of a trained PPO agent with and without RAE. The baseline agent is indifferent to the path to the goal, while the agent benefitting from RAE learns to follow the road, avoiding irreversible consequences.

### 6.3 Sokoban

Sokoban is a popular puzzle game where a warehouse keeper (controlled by the player) must move boxes around and place them in dedicated places. Each level is unique and involves planning, since there are many ways to get stuck. For instance, pushing a box against a wall is often un-undoable, and prevents the completion of the level unless actually required to place the box on a specific target. Sokoban is a challenge to current model-free RL algorithms, as advanced agents require millions of interactions to reliably solve a fraction of levels [46, 20]. One of the reasons for this is tied to exploration: since agents learn from scratch, there is a long preliminary phase where they act randomly in order to explore the different levels. During this phase, the agent will lock itself in unrecoverable states many times, and further exploration is wasted. It is worth recalling that contrary to human players, the agent does not have the option to reset the game when stuck. In these regards, Sokoban is a great testbed for reversibility-aware approaches, as we expect them to make the exploration phase more efficient, by incorporating the prior that irreversible transitions are to be avoided if possible, and by providing tools to identify such transitions.

We benchmark performance on a set of 1k levels. Results are displayed in Fig. 6. Equipping an IMPALA agent [15] with RAE leads to a visible performance increase, and the resulting agent consistently solves all levels from the set. We take a closer look at the reversibility estimates and show that they match the ground truth with high accuracy, despite the high imbalance of the distribution (*i.e.* few irreversible transitions, see Fig. 6c) and complex reversibility estim ation (see Fig. 6a).

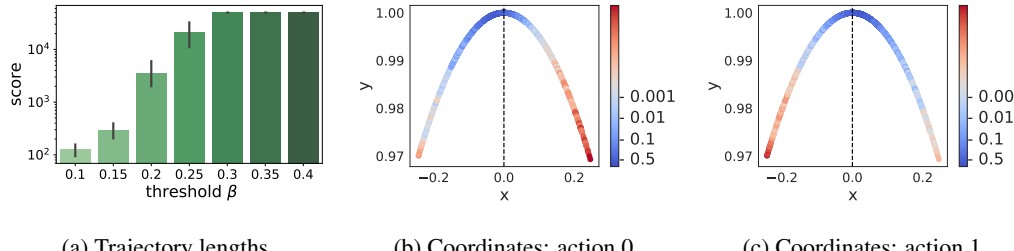

(a) Trajectory lengths      (b) Coordinates: action 0      (c) Coordinates: action 1

Figure 7: **(a):** Mean score of a random policy augmented with RAC on Cartpole+ for several threshold values, with 95% confidence intervals over 10 random seeds (log scale). **(b) and (c):** The $x$ and $y$ axes are the coordinates of the end of the pole relatively to the cart position. The color indicates the estimated reversibility values.

## 6.4 Safe Control

In this section, we put an emphasis on RAC, which is particularly suited for safety related tasks.

**Cartpole+.** We use the standard Cartpole environment, except that we change the maximum number of steps from 200 to 50k to study long-term policy stability. We name this new environment Cartpole+. It is substantially more difficult than the initial setting. We learn reversibility offline, using trajectories collected from a random policy. Fig. 7a shows that a random policy augmented with RAC achieves seemingly infinite scores. For the sake of comparison, we indicate that a DQN [31] and the state-of-the-art M-DQN [45] achieve a maximum score of respectively 1152 and 2801 under a standard training procedure, described in Appendix C.5. This can be surprising, since RAC was only trained on random thus short trajectories (mean length of 20). We illustrate the predictions of our learned estimator in Fig. 7b,7c. When the pole leans to the left ($x < 0$), we can see that moving the cart to the left is perceived as more reversible than moving it to the right. This is key to the good performance of RAC and perfectly on par with our understanding of physics: when the pole is leaning in a direction, agents must move the cart in the same direction to stabilize it.

**Turf.** We now illustrate how RAC can be used for safe online learning: the implicitly safe constraints provided by RAC prevent policies from deviating from safe trajectories. This ensures for example that agents stay in recoverable zones during exploration.

We learn the reversibility estimator offline, using the trajectories of a random policy. We reject actions whose reversibility is deemed inferior to $\beta = 0.2$, and train a PPO agent with RAC. As displayed in Fig. 8, PPO with RAC learns to reach the goal without causing any irreversible side-effect (*i.e.* stepping on grass) during the whole training process.

The threshold $\beta$ is a very important parameter of the algorithm. Too low a threshold could lead to overlooking some irreversible actions, while a high threshold could prevent the agent from learning the new task at hand. We discuss this performance/safety trade-off in more details in Appendix. C.7.

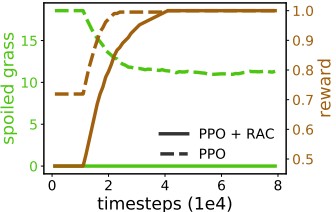

Figure 8: PPO and RAC (solid lines) vs PPO (dashed lines). At the cost of slower learning (brown), our approach prevents the agent from producing a single irreversible side-effect (green) during the learning phase. Curves are averaged over 10 runs.

## 7 Conclusion

In this work, we formalized the link between the reversibility of transitions and their temporal order, which inspired a self-supervised procedure to learn the reversibility of actions from experience. In combination with two novel reversibility-aware exploration strategies, RAE for directed exploration and RAC for directed control, we showed the empirical benefits of our approach in various scenarios, ranging from safe RL to risk-averse exploration. Notably, we demonstrated increased performance in procedurally-generated Sokoban puzzles, which we tied to more efficient exploration.

**Broader impact and ethical considerations.** The presented work aims at estimating and controlling potentially irreversible behaviors in RL agents. We think it has interesting applications in safety-first scenarios, where irreversible behavior or side-effects are to be avoided. The societal implication of these effects would be safer interactions with RL-powered components (*e.g.* robots, virtual assistants, recommender systems) which, though rare today, could become the norm. We argue that further research and applications should verify that the induced reversible behavior holds in almost all situations and does not lead to unintended effects. Our method could be deflected from its goal and used to identify and encourage actions with irreversible effects. In this case, a counter measure consists in using our method to flag and replace irreversible actions. Hence, while the method provides information that could be used to deal irreversible harm, we argue that it provides equal capabilities for detection and prevention.

## Acknowledgments and Disclosure of Funding

Experiments presented in this paper were partially carried out using the Grid'5000 testbed, supported by a scientific interest group hosted by Inria and including CNRS, RENATER and several Universities as well as other organizations. NG is a recipient of PhD funding from the AMX program, Ecole Polytechnique. The authors would like to thank Edouard Leurent, Antoine Moulin, Odalric-Ambrym Maillard, Léonard Hussenot, Nino Vieillard, Alexis Jacq, Théophane Weber and Bobak Shahriari for helpful comments and suggestions.

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
