# OpenReview forum: "There Is No Turning Back: A Self-Supervised Approach for Reversibility-Aware Reinforcement Learning"
_NeurIPS.cc/2021/Conference — NeurIPS 2021 Poster_

### Official Review · Reviewer_M8wY · 2021-06-28

**Rating:** 8
**Confidence:** 3

**Summary:**

The paper describes a way of adding a new inductive bias to RL algorithms, namely "taking irreversible actions is bad". The notion of reversibility is defined formally, together with a process for estimating it from samples. Experimentally, it is shown that using the new inductive bias helps achieving improved return on a number of toy benchmarks as well as on Sokoban.

**Limitations And Societal Impact:**

The authors provide a brief description of broader impact in lines 328-334. However, they do not currently envisage any clearly negative effects of their technology. One such effect that might be considered is that the technology might serve an an enabling factor in training agents designed to do irreversible harm. A discussion would be helpful.

**Main Review:**

The paper is highly original, high quality and very clearly written. I have no doubt that it is a good match for the NeurIPS community. In terms of significance, I rate it as above the bar. Also, I am really glad to see a piece of work that does away with the standard RL assumption that the MDP is ergodic.

My minor concerns are listed below:
1. What about stochastic MDPs? The benchmarks only seem to cover MDPs with deterministic transitions. It would be great to have at least one benchmark with stochastic transitions (to see how the self-supervised procedure performs, depending on the level of stochasticity).
2. The paper misses a link to literature (for example, [1]) on the region of attraction (ROA). ROA is the part of the state space where a controller can bring the system back to an equilibrium point (which is considered "safe"). Staying within ROA is somewhat analogous to following transitions that are easy to reverse. A discussion of this link would be helpful.
3.  How would you go about estimating $\phi(s,a)$? In Theorem 2, the paper includes a way to upper bound $\phi_{\pi}(s,a)$ (for a fixed policy), but it does not discuss how to do the supermum among policies.
4. Standard deviation across seeds is missing from Figure 6b (contrary to what you say in item 3c of the checklist).

Despite these, I am confident in the quality of the work. I will argue for the paper to be accepted.

Very minor:
- Figure 6 currently has no subcaptions.
- Figure 9 (in the appendix), mentions "Proposition 4" but refers to point 4 in the list to the left of the figure
- Line 502 (in the appendix): depends -> depend

[1] https://arxiv.org/pdf/1603.04915.pdf

**Time Spent Reviewing:**

4

---

> ### Author Response · Authors · 2021-08-10
> **Official response to Reviewer M8wY**
>
> We thank the reviewer for their positive assessment and insightful remarks.
>
> R: “What about stochastic MDPs? The benchmarks only seem to cover MDPs with deterministic transitions. It would be great to have at least one benchmark with stochastic transitions (to see how the self-supervised procedure performs, depending on the level of stochasticity).”
>
> A:
>
> This is a great point.
> To study this aspect, we use a 2d cliff walking gridworld where stochasticity comes from the wind: additionally to its move, the agent is pushed towards the cliff with a fixed probability. The agent gets a +1 reward for each timestep it stays alive, with a maximum of 250 timesteps.
> A reversibility-aware agent with a well calibrated threshold should avoid most moves that push it towards the cliff.
> We provide two tables: one with the average scores of a random policy and the other with the average scores of a random policy equipped with RAC. Rows correspond to varying stochasticity and columns to varying threshold values:
>
> Random policy:
>
> | p \ threshold | 0.   | 0.1  | 0.2  | 0.3  | 0.4  |
> |---------------|------|------|------|------|------|
> | 0.            | 57.5 | 57.7 | 61.2 | 58.2 | 57.7 |
> | 0.1           | 29.8 | 28.8 | 29.5 | 30.2 | 29.6 |
> | 0.2           | 18.6 | 18.5 | 19.3 | 18.9 | 18.8 |
> | 0.3           | 13.4 | 13.3 | 13.9 | 13.6 | 13.4 |
> | 0.4           | 10.5 | 10.7 | 10.4 | 10.2 | 10.2 |
>
> Random policy + RAC:
>
> | p \ threshold | 0.   | 0.1   | 0.2   | 0.3   | 0.4   |
> |---------------|------|-------|-------|-------|-------|
> | 0.            | 59.1 | 250.0 | 250.0 | 250.0 | 250.0 |
> | 0.1           | 29.2 | 56.0  | 56.3  | 80.2  | 248.5 |
> | 0.2           | 18.7 | 26.7  | 29.2  | 85.8  | 238.6 |
> | 0.3           | 13.2 | 16.8  | 19.6  | 77.6  | 250.0 |
> | 0.4           | 10.4 | 12.5  | 24.9  | 152.2 | 250.0 |
>
> We can notice several things:
> * a well-tuned threshold value is crucial to get decent performance
> * the optimal threshold increases with the stochasticity of the environment (but seems to quickly converge)
>
> R: “The paper misses a link to literature (for example, [1]) on the region of attraction (ROA). ROA is the part of the state space where a controller can bring the system back to an equilibrium point (which is considered "safe"). Staying within ROA is somewhat analogous to following transitions that are easy to reverse. A discussion of this link would be helpful.”
>
> A: Thanks for bringing ROA to our attention. Although the formalisms are different (Markov Decision Processes in the case of RL and nonlinear, continuous-time system in the case of continuous control), we indeed believe that the general goals are close: detecting which parts of the state space allow to go back to a “safe” zone, and using this information to explore safely. A major difference between the two lines of work are the assumptions regarding the environment model. Computing the ROA typically involves (full) prior knowledge about the environment dynamics. [1] is interesting in the sense that it only supposes the dynamics to be known up to a small residual, which is iteratively estimated from experiments on a real system. However, this is still in contrast with our approach, which is amenable in the model-free RL setting. These similarities and differences make such a discussion an interesting addition to the paper. We will discuss the link to ROA in the updated related work.
>
> R: “How would you go about estimating \phi(s, a) ? In Theorem 2, the paper includes a way to upper bound \phi_\pi(s, a) (for a fixed policy), but it does not discuss how to do the supermum among policies.”
>
> A:
>
> We agree that for some applications we might ultimately be interested in the general irreversibility $\phi(s, a)$, even if in our experiments we find $\phi_\pi$ to be sufficient.
>
> A relatively costly way to approximate $\phi(s, a)$ would be to keep track of successive checkpoints of $\phi_\pi$ along the training procedure and have $\tilde{\phi}(s, a) = \max_i \{ \phi_{\pi_i}(s, a) \}$. Another approximate approach would consist in keeping a large replay buffer with all previous interactions from various policies, and learning $\phi_\pi(s, a)$ with $\pi$ the mixture of policies that generate the replay buffer.
>
> We leave the study of these for future work.
>
> It can also be noted that Prop 1 makes a connexion between $\bar{\phi}_\pi$ and $\phi_K$ for arbitrary values of K, under some assumptions concerning $\pi$. As $\phi_K$ approximates $\phi$ when K is high, it can be seen as a theoretical way to link $\bar{\phi}_\pi$ and $\phi$.
>
> R: “Standard deviation across seeds is missing from Figure 6b (contrary to what you say in item 3c of the checklist).”
>
> A: Thanks for spotting this! This is an omission.
> Interestingly, the standard deviation for the Impala+RAE runs is quite inferior to that of the Impala baseline (for instance, asymptotically, std of ~2 for Impala+RAE against std of ~9 for Impala), which implies that RAE reduces performance variance.
> We will include standard deviation as shaded areas in the updated figure.
>
> R: “Very minor:
> * Figure 6 currently has no subcaptions.
> * Figure 9 (in the appendix), mentions "Proposition 4" but refers to point 4 in the list to the left of the figure
> * Line 502 (in the appendix): depends -> depend”
>
> A: Thanks for carefully checking the paper! We will fix these items in the updated draft.
>
> R: “The authors provide a brief description of broader impact in lines 328-334. However, they do not currently envisage any clearly negative effects of their technology. One such effect that might be considered is that the technology might serve as an enabling factor in training agents designed to do irreversible harm. A discussion would be helpful.”
>
> A: This is an interesting aspect that we agree should be discussed. One could envision a mitigation strategy where the $\phi$ component of RAC is used to flag potentially irreversible actions on top of the AI and either a) our proposed rejection sampling mechanism, b) a human operator or c) a default, known-to-be-safe policy overrides those if necessary. Thus, while the method provides information that could be used to deal irreversible harm, we argue that it provides equal capabilities for detection and prevention.
> We will integrate these elements in the updated draft.

---

### Official Review · Reviewer_XsJi · 2021-07-14

**Rating:** 6
**Confidence:** 4

**Summary:**

This work proposes reversability awareness during training. The agent is augmented with an intrinsic reward / constraint which deters it from taking irreversible actions.

**Limitations And Societal Impact:**

-

**Main Review:**

The concept of irrevesibility seems very important especially in the context of robotics. The behavior of avoiding irreversible behavior can be linked to human behavior and how we try to maintain a steady state in the environment.

Of course, this method isn't some golden bullet for RL and there are many cases where one would want reversability or where it could help.

My only remark would be that I'd like the authors to also present some failure cases. Clearly, reversability can lead to unwanted results. Either slightly sub-optimal behavior or when the weight of the reversability reward is too high, it can lead to utter failure.

**Time Spent Reviewing:**

3

---

> ### Author Response · Authors · 2021-08-10
> **Official response to Reviewer XsJi**
>
> We thank the reviewer for their comments.
>
> R: “Of course, this method isn't some golden bullet for RL and there are many cases where one would want reversibility or where it could help.”
>
> A:
>
> Indeed, some tasks require irreversible actions to be performed in order to succeed. Hence, avoiding irreversible actions can be a good strategy in some cases, but also lead to suboptimal behavior in others. As we mentioned in our response to RJAjY, it is very important to notice that while RAC avoids all actions flagged as irreversible, RAE only induces a *bias* towards reversible actions, but the agent can still choose irreversible actions if the benefits outweigh the costs. This effect is most visible in Sokoban, where approximately half of the levels cannot be solved without performing at least one irreversible action (e.g. pushing a box on a wall). Impala+RAE manages to solve close to all the tasks, which clearly indicates that the method leaves room for irreversible actions to be taken.
>
> Alternatively, our methods could be adapted to the opposite setting where one would want to favor irreversible moves: using RAE, by rewarding irreversible transitions instead of penalizing them, or using RAC, by rejecting reversible actions.
>
> Finally, we would like to emphasize the fact that the notion of reversibility is often useful in safety settings, where maximizing the reward is not the only goal. In Section 6.4, we illustrate the usefulness of the method in two experiments, which show that RAC can be useful to ensure that the agent stays in a safe zone (Cartpole+) or to learn a policy while minimizing irreversible side-effects (Turf).
>
> R: “My only remark would be that I'd like the authors to also present some failure cases. Clearly, reversibility can lead to unwanted results. Either slightly sub-optimal behavior or when the weight of the reversibility reward is too high, it can lead to utter failure.”
>
> A:
>
> We agree with the line of reasoning of the reviewer. In fact, we actually provide some failure cases in the paper.
>
> Some can be found in Appendix C.7, where we highlight the trade-off between performance and avoiding irreversibility in Turf. We can see in Fig. 11 that both the sample-efficiency and the number of irreversible side-effects decreases as the threshold increases. Too high a threshold (0.4) prevents the agent from reaching the goal at all by fear of commiting irreversible actions, which is a clear failure case.
>
> Others can be found in Fig. 7a (Cartpole+), where too low a threshold results in visibly reduced performance compared to higher, more suitable threshold values.
>
> As the reviewer mentions, the $\beta$ threshold is important to set right to get the best of our method.
>
> In Appendix C.7, we advocate for a careful tuning of the threshold when computationally feasible: it should be initialized at a high value (e.g. 0.5) and decreased progressively, until the desired agent behaviour is reached. This procedure would ensure that the chosen threshold is the maximal threshold such that the environment can still be solved.

---

### Official Review · Reviewer_JAjY · 2021-07-14

**Rating:** 5
**Confidence:** 3

**Summary:**

The paper defines several reversibility measures in MDPs and provides corresponding results for estimating those measures. Then, a supervised learning approach is proposed to be incorporated into an RL agent for avoiding irreversible actions.

**Limitations And Societal Impact:**

Not relevant.

**Main Review:**

The idea of inspecting reversibility is itself novel and I appreciate the fresh POV. However, I'm not convinced the approach proposed in the paper is indeed viable.

The initial definitions and results are quite clear and intuitive. The theorems are "nice to have", but they do not convey any guarantees or insights regarding the algorithm itself. As the reading progresses, I've had a hard time following what exactly is proposed. From what I understand, the irreversibility prediction is used as negative intrinsic reward for discouraging visitations of such states.

My main issue with the paper is the following. Training the predictor on multiple pairs of states feels like a very inefficient way of avoiding certain areas of the state-space. Informally, a task with complexity O(n^2) is conducted while an O(n) disastrous-state identification might be just a good. The latter idea has been explored quite a lot and a comparison is warranted. Considering the cartpole example, simply identifying the states at which the pole is close to falling might have given similar outcomes at a much smaller computational price.

All in all, while I agree with intuition that the human mind is suspicious and attentive to irreversible actions, I think its implication on an RL agent should be more intricate than simply trying to avoid such decisions.

***Post-rebuttal response:***
My main concern is the O(n^2) issue and I have a hard time accepting the authors' answer to it. In a sense, they are saying that this is not really a problem because i) the setting is not tabular but of a NN; and ii) it works well on an example. I disagree with claim i) and don't see how it answers my question. I don't think complexity claims can be waived by referring to NNs as being magical. As for the Sobokan experiment -- it is of course better to have than not, but it's not enough as a sole argument. I still lean towards rejection.

**Time Spent Reviewing:**

3

---

> ### Author Response · Authors · 2021-08-10
> **Official response to Reviewer JAjY**
>
> We thank the reviewer for their comments. We provide additional remarks and clarifications in what follows.
>
> R: “The idea of inspecting reversibility is itself novel and I appreciate the fresh POV.”
>
> A: Thanks!
>
> R: “The theorems are "nice to have", but they do not convey any guarantees or insights regarding the algorithm itself.”
>
> A:
>
> Theorem 2 and Proposition 1 are key points of our work. They both show that the empirical reversibility, which is the quantity our methods estimate in this work, cannot be arbitrarily low if the action is actually reversible. In particular, Theorem 2 is a novel theorem that has very practical consequences. It is what motivates the use of precedence to approximate reversibility.
>
> Given a dataset of trajectories from a behavior policy with sufficient coverage of the state space, and a sufficient amount of training, the approximation error of the learned empirical reversibility should get close to zero with a large probability.
>
> R: “As the reading progresses, I've had a hard time following what exactly is proposed.”
>
> A: We would be happy to improve the clarity of the writing.
> We welcome additional suggestions in that direction, but we kindly note that other reviewers (RM8wY, RYdib) found the writing clear.
>
> R: “From what I understand, the irreversibility prediction is used as negative intrinsic reward for discouraging visitations of such states.”
>
> A: This is correct, though incomplete. RAE indeed penalizes irreversible transitions. RAC, instead, avoids irreversible actions entirely by rejecting action samples whose empirical reversibility is lower than a chosen threshold.
>
> R: “Training the predictor on multiple pairs of states feels like a very inefficient way of avoiding certain areas of the state-space. Informally, a task with complexity O(n^2) is conducted while an O(n) disastrous-state identification might be just a good. The latter idea has been explored quite a lot and a comparison is warranted.”
>
> A:
>
> We argue that the complexities mentioned are not accurate outside the tabular case. They hide the potential that neural network approximators have regarding generalization, which is what motivates our approach in the first place. Typically, the precedence classifier has incentive to learn features that are predictive of the arrow of time and can generalize.
>
> We exemplify this in the Sokoban task: while the number of state pairs is very large, we observe empirically that the precedence classifier manages to learn that all actions that push a box in a corner cannot be reversed, and does so after having seen only a tiny fraction of all such state pairs.
>
> A notable advantage of our approach is that it does not rely on the knowledge of the reward function, so the precedence estimation transfers to tasks with similar dynamics but different reward functions.
>
> We would appreciate relevant pointers to the disastrous-state identification literature, which we are not familiar with, but would be happy to add. Informally, we fail to think of a state-only approach that would be learned in a self-supervised way, like the proposed method.
>
> R: “Considering the cartpole example, simply identifying the states at which the pole is close to falling might have given similar outcomes at a much smaller computational price.”
>
> A: In Cartpole, it is not enough to know which states are to be avoided to reproduce the safe interactions RAC provides *with arbitrary policies* (i.e. avoiding all irreversible transitions from the very first interaction in the environment). To accomplish that, one has to know which actions are irreversible in order to avoid them. While knowing which states are to be avoided might be enough when modifying reward functions (i.e. after the transition is done), it does nothing to guide action selection (i.e. before the transition is done).
>
> R: “All in all, while I agree with intuition that the human mind is suspicious and attentive to irreversible actions, I think its implication on an RL agent should be more intricate than simply trying to avoid such decisions.”
>
> A: We think that an important distinction is to be made here. RAE incentivizes the agent to select reversible actions, but the optimal action can still be irreversible as long as the benefits (i.e. the cumulative sum of future rewards) outweigh the costs (i.e. the cumulative sum of future irreversibility penalties plus the cumulative sum of future rewards of the best other action). A successful example for this kind of behavior is Sokoban, where we estimate approximately half of the generated levels to feature an irreversible action in every successful sequence of actions. The fact that Impala+RAE reaches an asymptotic average score of close to 10 (which is the best score in each level) proves that the agent indeed takes irreversible actions when needed. We will emphasize that point in the updated draft.

---

### Official Review · Reviewer_Ydib · 2021-07-14

**Rating:** 9
**Confidence:** 4

**Summary:**

This paper introduces a self-supervised way to measure the reversibility of events. This is done by learning a classifier that takes as input two states from a trajectory and predicts which came first. The paper provides strong theoretical motivation for using this as a measure of reversibility. They then run experiments that use this reversibility metric in two ways: Reversibility-Aware Exploration (RAE), which penalizes the agent with a negative reward when it does an action that is not reversible enough; and Reversibility-Aware Control (RAC), which uses rejection sampling to ensure that actions sampled from the policy are reversible.

In the experiments, it is shown that: (1) reversibility is a strong intrinsic reward, capable of training a Cartpole agent without access to the true rewards; (2) in Sokoban, using IMPALA+RAE results in improvements in sample efficiency; and (3) using a classifier learned using offline data, they show RAC is capable of avoiding irreversible side effects during the whole training process in a custom grid world environment as well as enabling a Cartpole agent with a random policy to perform optimally using only rejection sampling to avoid irreversible actions.

**Ethical Concerns:**

I have no ethical concerns with this paper.

**Limitations And Societal Impact:**

The authors provide a strong case for why their method can help avoid bad side effects when deploying RL systems.

**Main Review:**

1. Originality: This paper introduces a novel and simple approach for estimating the reversibility of events. The related work section is very detailed.
2. Quality: The quality of this paper is high. I appreciate the clear explanation of what exactly reversibility is and how it is related to classifying the temporal order of events. The experiments are well thought out, and I think the fact that using reversibility as an intrinsic reward in environments like Cartpole works as well as it does is very cool.
3. Clarity: I have no complaints with the clarity of this paper. It is well written and the visualizations and plots are intuitive.
4. Significance: Due to the simplicity and strength of this approach, I think this is an important contribution to both the exploration and AI safety fields.

**Time Spent Reviewing:**

5

---

> ### Author Response · Authors · 2021-08-10
> **Official response to Reviewer Ydib**
>
> We thank the reviewer for the positive feedback on the proposed method. We are glad that they consider our approach an important contribution.

---

### Decision · Program_Chairs · 2021-09-27

**Decision:**

Accept (Poster)

**Comment:**

After reading each other's reviews and the authors' feedback, the reviewers discussed the merits and flaws of the paper.
The reviewers did not reach a consensus about the acceptance of this paper.
In particular, the main concern is about the quadratic complexity of the proposed approach, which has not been properly addressed by the authors' answers. Nonetheless, the majority of the reviewers think that the good performance on the Sokoban experiment shows that the algorithm can be effective in practice. Overall I think that the proposed approach is interesting and I am in favor of its acceptance.
I want to congratulate the authors and invite them to modify their paper following the reviewers' suggestions.